# Should We Monitor Glucose and Biomarkers in Diabetics over Heart Surgery?

**DOI:** 10.3390/jcm10153399

**Published:** 2021-07-30

**Authors:** Elena Z. Golukhova, Ljubov S. Lifanova, Yaroslava V. Pugovkina, Marina V. Grigoryan, Naida I. Bulaeva

**Affiliations:** Bakulev National Medical Research Center of Cardiovascular Surgery, Cardiology Department, Roublyevskoe Shosse 135, 121552 Moscow, Russia; egolukhova@yahoo.com (E.Z.G.); ljubovlifanova93@mail.ru (L.S.L.); viktorovna_y87@mail.ru (Y.V.P.); m_grigoryan@mail.ru (M.V.G.)

**Keywords:** diabetes mellitus, insulin, glucose monitoring, coronary bypass surgery, outcomes

## Abstract

Hyperglycemia is associated with adverse outcomes after coronary artery bypass grafting (CABG). While there is a consensus that blood glucose control may benefit patients undergoing CABG, the role of biomarkers, optimal method, and duration of such monitoring are still unclear. The aim of this study is to define the efficacy of a continuous glucose monitoring system (CGMS) and link it to pro-inflammatory biomarkers while on insulin pump therapy in diabetic patients undergoing CABG. We prospectively assessed CGMS for 72 h in 105 patients including 52 diabetics undergoing isolated CABG. In diabetics, CGMS was connected to an insulin pump for precise glucose control. On top of conventional biomarkers (HbA1C, lipid profile), high sensitive C-reactive protein (hs-CRP), Regulated upon Activation Normal T cell Expressed and presumably Secreted (RANTES), and leptin levels were collected before surgery, 1 h, 12 h, 7 days, and at 1 year after CABG. Overall, CGMS revealed high glucose independently from underlying diabetes during first 48 h following CABG but was higher (*p* < 0.05) in diabetics. The insulin pump improved glycemic control over early follow-up (72 h) post-CABG. There were no hypoglycemic episodes in patients on insulin pump therapy and those receiving bolus insulin therapy. We revealed a lower rate of postpericardiotomy syndrome (PCTS) in patients on insulin pump therapy compared to patients prescribed bolus insulin therapy in the early postoperative period (*p* = 0.03). Hs-CRP and RANTES levels were lower in patients with T2DM on insulin pump therapy compared to patients prescribed bolus insulin therapy in the early postoperative period (*p* < 0.05). It is most likely due to the fact that insulin pump therapy decreases systemic inflammatory response. Further controlled trials should assess whether CGMS improves outcomes after cardiac surgery.

## 1. Introduction

Coronary artery disease (CAD) is the leading cause of death worldwide. It is well known that type 2 diabetes mellitus (T2DM) is one of the major cardiovascular risk factors and results in atherosclerotic plaque progression [1]. About 70% of patients with T2DM have coronary artery disease [2]. Nearly 50% of patients undergoing coronary artery bypass grafting (CABG) suffer from T2DM, which causes reduced graft patency and recurrent ischemic events [3,4,5,6]. However, in patients without T2DM, perioperative hyperglycemia is associated with negative outcomes after CABG. It is most likely that intensive insulin therapy is preferable in order to achieve the target glucose level. However, hypoglycemic events may develop due to aggressive glycemic control, which can also be harmful. It is currently unclear as to which approach leads to better perioperative glucose level control and reduced complication rates. Continuous glucose monitoring systems (CGMS) could prevent severe hyper- and hypoglycemia in the perioperative CABG period [7]. Moreover, continuous insulin infusion results in both glycemic control improvement and a reduced rate of complications in the perioperative CABG-period [8]. There is no data on the specific benefits of insulin pump therapy in patients undergoing CABG, and the prognostic value of various biomarkers is not well identified. We therefore analyzed common metabolic disorders perioperatively by comparing two groups: patients with DM2 (study group) and without DM2 (control group) to identify potential benefits and drawbacks of insulin pump therapy over CABG. Here, we validated the CGMS approach while on insulin pump therapy in diabetics undergoing CABG in intra- and early post-surgery, linking the poor outcomes with certain biomarkers.

## 2. Methods

The study was conducted in the Bakulev National Medical Research Center for Cardiovascular Surgery (2017–2019). We included 105 patients aged 61 to 81 years of age with documented CAD who underwent isolated CABG. Both on-pump and off-pump coronary surgeries were permitted. According to the local Ethics Committee protocol (№ 2; 21 March 2016), permission had been granted, and all patients signed agreement forms and informed consent. The main exclusion criteria were diabetes mellitus Type 1, hepatitis, recent stroke, psychiatric diseases, infections, cancer, iodine contrast allergy, pregnancy, and history of gastrointestinal bleeding. Patients who underwent CABG were divided into two groups: 52 diabetics, and 53 patients without T2DM. Both groups of patients were well matched in terms of their demographics, medical history, and intervention strategy (Table 1). 

Body mass index (BMI) in the patients with T2DM was higher: 30.5 kg/m^2^ vs. 28.7 kg/m^2^ in patients without T2DM (*p* = 0.03). We performed a physical examination of the patients, including commonly used blood tests (such as HbA1C, lipid profile, etc.), electrocardiography (Bioset 3500 ECG device, HORMANN Medizintechnik Zwonitz, Germany), echocardiography, a carotid and lower limb arteries duplex scan («Philips iE-33», Philips Medical Systems, Bothell, WA, USA), and coronary angiography (INTEGRIS-3000, Philips, Netherlands and ANGIOSCOP D-33, Siemens, Germany). We assessed glycemic profile and glycated hemoglobin level (HbA1c) before CABG. The target HbA1c level before CABG was less than 7%. Antihyperglycemic drugs were canceled 48 h for biguanides and 24 h for sulfonylureas before surgery, and the patient was prescribed basal-bolus insulin therapy under the supervision of the endocrinologist. Based on the 2017 European Association of Thoracic Surgeons (EACTS) Guidelines on perioperative medication in adult cardiac surgery, a target glucose level is between 8.3 and 10.0 mmol/L in the early post-CABG period. The hypoglycemia cut-off was defined as a glucose level that was less than 3.9 mmol/L. Serial measures of highly sensitive C- reactive protein (hs-CRP) levels were taken before the surgery and at 1 h, 12 h, 7 days, 1 year after CABG. RANTES (Regulated upon Activation Normal T cell Expressed and presumably Secreted) levels were detected by the immune enzyme assay (eBioscience, San Diego, CA, USA) before surgery and 7 days and 1 year after CABG. Normal ranges for RANTES levels were 37,025–101,161 pg/mL. Serum leptin concentration was also determined by the immune enzyme assay (Diagnostics Biochem Canada Inc, London, Ontario, Canada) before the surgery and7 days and 1 year after CABG. Normal ranges for serum leptin levels in women were 3.6–11.1 ng/mL and in men 2.0–5.6 ng/mL. In both groups of patients with and without T2DM, CGMS was used in the intra- and early postoperative periods (72 h). CGM sensors were inserted into the subcutis of the upper back and connected to transmitters. The interstitial fluid glucose levels were measured every 5 min. The transmitters wirelessly sent information to the pump monitors. To control perioperative glycaemia, we used two methods: the insulin pump system (MiniMed Paradigm Veo 554/754) and the standard hospital protocol in the early postoperative period (72 h). We then compared the results. Patients with T2DM were randomly divided into two subgroups. In the first subgroup (24 patients), CGMS was associated with insulin pump therapy for successful postoperative glucose control; in the second subgroup, intravenous insulin injections were prescribed according to the intravenous blood glucose measurements (BECKMAN Synchron CX7 Synchron-CX7, Brea, California, USA) every 3 h according to the hospital standard protocol. CGMS data were used to estimate blood glucose.

In patients without T2DM, the hospital standard protocol applied. Primary outcomes of the study outlined before enrollment were length of stay, all-cause death, and complications developed within 7 days, such as postpericardiotomy syndrome (PCTS), pneumonia, deep sternal wound infection (DSWI), sepsis, renal failure, acute coronary syndrome, stroke, arrhythmia. We also analyzed major adverse cardiovascular events such as acute coronary syndrome (ACS), stroke, death, and repeat revascularization in both subgroups of patients with T2DM at 1 year post-CABG. We interviewed all patients and invited them for examination at 1 year post-CABG. Blood tests (hs-CRP, RANTES, leptin, adiponectin), electrocardiography, echocardiography, and waist circumference measurements were completed in the patients who visited us. We performed a coronary angiography in patients with a positive stress ECG test. 

### Statistical Analysis

The data were analyzed with SPSS Statistics 21 software. The results were presented through descriptive statistics (median, first and third quartiles) for the numerical variables and through number and proportion for qualitative variables. The groups were compared through nonparametric methods (Mann–Whitney test, Fisher’s exact test). We also used the Wilcoxon signed-rank test to compare two related samples and the correlation analysis.

## 3. Results

There was a universal consistent high blood glucose level during the 48 h after CABG, according to CGMS in patients with and without T2DM, although this was higher in diabetics (*p* < 0.05) (Figure 1). Notably, the highest glucose level was recorded over first the 12 h post-CABG in both groups of patients. This fact suggested that there was a stress induced hyperglycemia as a protective response to CABG, however, severe hyperglycemia may be deleterious and often results in adverse outcomes after surgery. Based on the 2017 European Association of Thoracic Surgeons (EACTS) Guidelines, we chose a target glucose level between 8.3 and 10.0 mmol/L in the early post-CABG period. In 62.5% of patients with T2DM on insulin pump therapy, this target glucose level was achieved during the 12 h after the surgery. In contrast, only in 21.4% of diabetics prescribed bolus insulin therapy achieved the target glucose level in the early follow-up frame (*p* = 0.003). 

There were no hypoglycemic episodes (the glucose level less 3.9 mmol/L) in the patients with T2DM in both subgroups, which proved that both the insulin pump therapy and bolus insulin therapy were safe in the early follow-up periods. (Figure 2). There was a significant decrease in glucose levels according to the CGMS during the first 12 h after the surgery in patients on insulin pump therapy (*p* = 0.022). In contrast, in patients on bolus insulin therapy, there was a significant increase in glucose levels according to the CGMS during the 12 h after the surgery (*p* = 0.033). The glucose level in patients on insulin pump therapy was lower compared to patients with prescribed bolus insulin therapy in early post-CABG (*p* = 0.021). However, there was no significant difference in the glucose levels in both subgroups of patients with T2DM on the 2nd and 3rd days of the post-CABG period. There was a gradual glucose decrease without sharp fluctuations in patients on insulin pump therapy compared to patients on bolus insulin therapy (Figure 2).

There was significant increase in the hs-C-reactive protein in both subgroups of patients with T2DM during the 12 h after the surgery (*p* = 0.0001 for both). The hs-C-reactive protein level in patients on insulin pump therapy was lower compared to the patients prescribed bolus insulin therapy in the early postoperative period (during 12 h after the surgery) (*p* = 0.0001). However, there was no significant difference in the hs-C-reactive protein levels in both subgroups of patients with T2DM on the 7th day and 1 year after the post-CABG period (Figure 3). 

We also determined that the RANTES level in patients on insulin pump therapy was lower compared to patients prescribed bolus insulin therapy in the early postoperative period (on the 7th day after the surgery in patients on insulin pump therapy—1200 (1075.0; 1700.0); in patients on bolus insulin therapy—1800.0 (1700.0; 22,850.6) pg/mL; (*p* = 0.035)). However, there was no significant difference in RANTES levels in both subgroups of patients with T2DM after 1 year of CABG. Strong positive correlations were found between the RANTES, the hs-CRP levels, and the rate of postcardiotomy syndrome (PCTS) in the early postoperative period (r = 0.74; *p* = 0.023; r = 0.4; *p* = 0.012). There were no dramatic changes in the rate of complications such as acute coronary syndrome (ACS), stroke, infections (pneumonia, mediastinitis, sepsis), acute renal failure in both subgroups of patients with T2DM in the early postoperative period. There were no deaths in either of the subgroups of patients with T2DM. However, we observed a decreased rate of postcardiotomy syndrome (PCTS) in the patients on insulin pump therapy via MiniMed Paradigm Veo 554/754 compared to prescribed bolus insulin patients in the early postoperative period (*p* = 0.03) (Figure 4). 

There was a significant decrease in the length of stay in patients on insulin pump therapy compared to patients on bolus insulin therapy (on insulin pump therapy—9.0 (8.25; 10.75); on bolus insulin therapy—12.0 (10.0; 14.0) days; *p* = 0.007) due to the lower rate of postcardiotomy syndrome in patients on insulin pump therapy (r = 0.63; *p* = 0.0001). There were no significant differences in the rate of complications such as acute coronary syndrome (ACS), stroke, repeat revascularization, or death in either of the subgroups of patients with T2DM at 1-year post-CABG. 

We also analyzed biomarkers of metabolic disorders. According to our analysis, we found a decreased leptin level in patients on insulin pump therapy compared to patients prescribed bolus insulin therapy in the early postoperative period (on the 7th day after the surgery in patients on insulin pump—16.25 (12.03; 35.65); in patients on bolus insulin therapy—54.3 (49.7; 60.9) ng/mL; *p* = 0.039). However, there was no significant difference in the leptin level in either of the subgroups of patients with T2DM at 1-year post-CABG. A strong positive correlation between the leptin level and the rate of postcardiotomy syndrome (PCTS) was observed early post-CABG (r = 0.725; *p* = 0.027). 

## 4. Discussion

We found that CGMS is a reliable, safe, and efficient tool to monitor insulin pump usage in diabetics undergoing CABG. The novelty of the index study is the hint that certain biomarkers may be linked to worsened CABG outcomes. Indeed, according to the 2019 ESC Guidelines on diabetes, severe hypoglycemia is associated with an increased risk of death and poor cardiovascular prognosis [9]. Therefore, it is necessary to avoid hypoglycemia (class I, level C) at almost any cost. It is well established that continuous glucose monitoring systems are valuable tools to improve glycemic control (class IIa, level A), and they can also prevent severe hyper- and hypoglycemia over CABG. Insulin pump therapy is the one of methods used to achieve desirable glucose targets. What are the advantages and disadvantages of insulin pumps? Despite the fact that insulin pump therapy was not common approach in patients undergoing CABG, we decided to assess the validity and efficacy of this technique. Based on the index findings, there was a high glucose level reported during the 48 h after the surgery according to CGMS in patients with and without T2DM. It is well-known that perioperative hyperglycemia is associated with adverse outcomes after coronary artery bypass grafting [10]. Thus, successful perioperative blood glucose control is necessary to reduce the risk of death and postoperative complications. 

In our study the insulin pump system and the standard hospital protocol (insulin bolus therapy) were used at early post-CABG (72 h). The target glucose level (between 8.3 and 10.0 mmol/L) was more frequently achieved in patients with T2DM on insulin pump therapy compared to patients on bolus insulin therapy in the early post- CABG period. There were no hypoglycemic episodes in patients with T2DM in both subgroups, which proved that both insulin pump and bolus insulin therapies are both safe during early follow-up. Obviously, the use of continuous glucose monitoring systems allows the thorough analysis the glycemic profile and the avoidance of hypo- and hyperglycemia in the perioperative period. Additionally, the main advantage of this technique is the efficacy of insulin pump therapy. “Threshold Suspend” is the system that is integrated into the insulin pump that stops insulin infusion for 2 h until the glucose level is normal. The patient can resume insulin infusion himself. The disadvantage is the higher cost. Insulin pump therapy is used in the intra- and early postoperative periods of mini-invasive surgeries. Hypoglycemic episodes are less common in patients on insulin pump therapy compared to patients on insulin bolus therapy [11]. There are no data on insulin pump therapy in patients undergoing CABG. Further studies are needed to optimize perioperative glucose management. 

It should be noted that there was a decrease of hs-C-reactive protein and RANTES levels in patients on insulin pump therapy compared to the bolus insulin therapy group during the first 7 days after the surgery. Obviously, both the hs-C-reactive protein and RANTES are biomarkers of inflammation. However, hs-CRP is considered to be “an acute phase protein”, while RANTES is the activator of chronic inflammation. RANTES is a chemokine that is responsible for the influx of blood cells, including T and B lymphocytes, monocytes, neutrophils, eosinophils, and basophils. The insulin pump therapy was most likely reducing systemic inflammatory response syndrome (SIRS), which is associated with a lower rate of infections in patients on insulin pump therapy at early post-CABG. As for the incidence of delayed complications in patients with DM2 at 1-year post-surgery, there were no significant differences. 

Adequate glycemic control is necessary to reduce post-CABG complications. Lorusso R. et al. considered the poor glycemic management in patients with T2DM to be associated with atherosclerosis progression and worse graft patency [7]. According to the 2017 European Association of Thoracic Surgeons Guidelines, there is no standard approach to manage glycemic control, while the results of different studies are controversial [8]. To choose the target glycemic level is important. The problem is finding a balance between the effective prevention of post-CABG complications without provoking potential hypoglycemia. The Controlled Trial of Intensive Versus Conservative Glucose Control in Patients Undergoing Coronary Artery Bypass Graft Surgery (GLUCO-CABG) study showed that the use of intensive insulin therapy to achieve the target glucose level between 100 and 140 mg/dl in the ICU did not significantly decrease the rate of perioperative complications compared to the target glucose level being between 141 and 180 mg/dl after CABG [12]. The Normoglycemia in Intensive Care Evaluation and Surviving Using Glucose Algorithm Regulation (NICE-SUGAR) trial demonstrated that a blood glucose level between 81 and 108 mg/dl was associated with a significant increase in all-cause mortality in ICU patients compared to a target of 180 mg/dl or less [12]. The comparison of intensive and moderate glucose controls was beyond the reach of this study. However, based on our results there was no dramatic increase in the rate of post- CABG complications such as PCTS, pneumonia, deep sternal wound infection, sepsis, renal failure, acute coronary syndrome, stroke, arrhythmia, and length of stay in patients with moderate glucose control (8.3–10.0 mmol/L). 

Importantly, patients on insulin pump therapy achieved the target glycemic level more frequently than patients treated with bolus insulin therapy. Additionally, the insulin pump therapy approach leads to the easier adjustment of insulin dosing and hyperinsulinemia prevention. 

There was a significant positive correlation between the leptin level and the insulin dosage. By affecting the adipocyte metabolism, intravenous insulin infusion stimulates leptin production [13]. In our study, we revealed the decreased leptin level in patients on insulin pump compared to the patients with prescribed bolus insulin therapy at early post-CABG. Thus, it seems that insulin pump therapy prevented excessive insulin concentrations. High leptin increases the synthesis of pro-inflammatory cytokines, promotes the development of insulin resistance, and predisposes the patient to metabolic and vascular complications [14], while low leptin may decrease the rate of PCTS. Additionally, according to our results, the average waist circumference was increased at 1-year post-CABG, and there was a strong positive correlation between the leptin level and waist circumference in patients with T2DM at 1-year post-CABG. Perhaps this trend is associated with sedentary lifestyle or fast food consumption. Therefore, it is necessary to promote a healthy lifestyle, including healthy eating, physical activity, weight and stress management in patients after CABG. 

### Limitations and Strengths

Several limitations of this study should be mentioned. First, the sample size was rather small, and the study was conducted for over 2 years of enrollment. The prospective rather than a randomized design is also a shortcoming. We did not conduct the study exclusively in diabetics, and other biomarkers may offer some advantage in glucose monitoring assessment. We divided the subgroups based on pump use (on-, or off), which may potentially be avoided in the future studies. Finally, we used non-diabetics as a control group, limiting reliable comparisons even further. Nonetheless, the strengths of our study were the uniformed enrollment rules, the single-center setup, and the considerable follow-up duration, which lasted for at least 12 months, resulting in the collection broad outcomes. 

## 5. Conclusions

Consistent glycemic control is important post-CABG and allows the imrpovement of outcomes and the reduction of the risk of postoperative complications whether or not the patient is diabetic. Insulin pump therapy allows the improvement of glycemic control and can potentially reduce the risk of the most common complication after CABG—postpericardiotomy syndrome. Indeed, insulin pump therapy is a reliable, safe and efficient tool for glucose control in diabetics undergoing CABG. Further controlled trials should assess whether CGMS improves clinical outcomes after cardiac surgery.

## Figures and Tables

**Figure 1 jcm-10-03399-f001:**
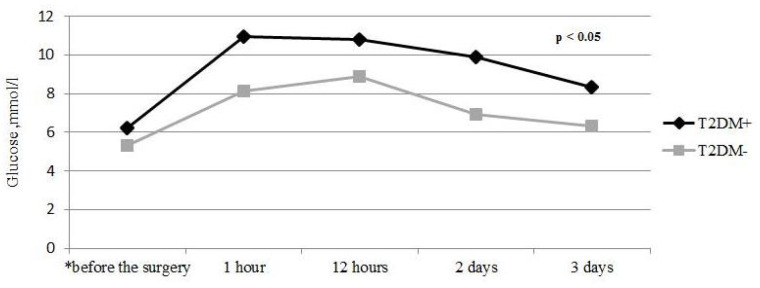
Glucose levels dynamics in patients with and without T2DM in the perioperative period according to CGMS, mmol/L. * Glucose levels in patients with T2DM (Ме (Q25; Q75)) before the surgery—6.02 (5.5; 8.28); 1 h after the surgery—10.95 (9.48; 13.2); 12 h after the surgery—10.8 (9.14; 13.28); 2 days after the surgery—9.9 (8.08; 12.73); and 3 days after the surgery—8.35 (6.73; 10.0) mmol/L. Glucose levels in patients without T2DM (Ме (Q25; Q75)) before the surgery—5.3 (4.9; 6.19); 1 h after the surgery—8.15 (7.2; 9.4); 12 h after the surgery—8.9 (7.18; 11.2); 2 days after the surgery—6.95 (6.0; 8.23); and 3 days after the surgery—6.3 (5.6; 8.1) mmol/L.

**Figure 2 jcm-10-03399-f002:**
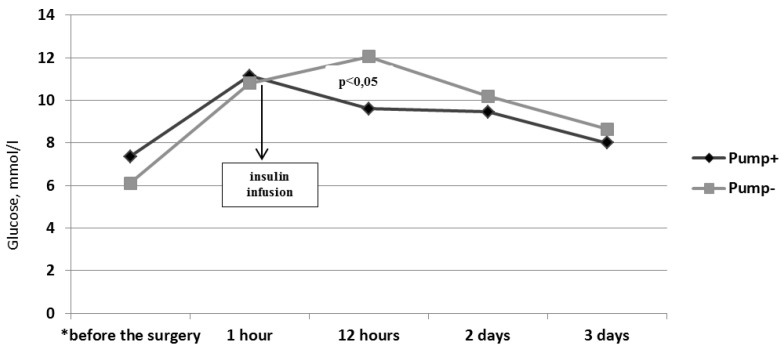
Glucose levels dynamics in patients with T2DM in the perioperative period according to CGMS, mmol/L. * Glucose levels in patients on insulin pump (Pump+) (Ме (Q25; Q75)) before the surgery—7.35 (5.63; 9.15); 1 h after the surgery—11.5 (9.78; 15.07); 12 h after the surgery—9.6 (8.95; 11.53); 2 days after the surgery—9.45 (7.8; 10.48); and 3 days after the surgery—8.0 (6.28; 9.89) mmol/L. Glucose levels in patients on bolus insulin therapy (Punp-) (Ме (Q25; Q75)) before the surgery—6.1 (5.05; 6.88); 1 h after the surgery—10.8 (9.0; 12.65); 12 h after the surgery—12.05 (10.13; 14.65); 2 days after the surgery—10.2 (8.39; 13.025); and 3 days after the surgery—8.65 (7.53; 10.28) mmol/L.

**Figure 3 jcm-10-03399-f003:**
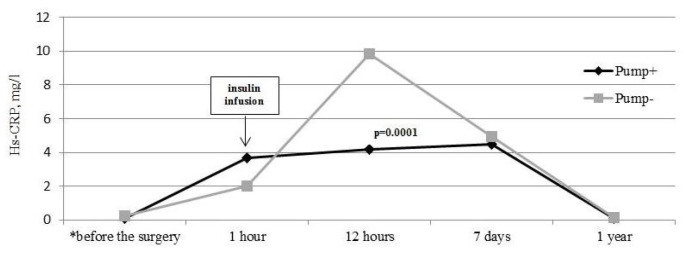
Hs-C-reactive protein (CRP) levels in patients with T2DM in the perioperative period, mg/L. * Hs-CRP levels in patients on insulin pump (Pump+) before the surgery—0.09 (0.066; 0.288); 1 h after the surgery—3.67 (1.2; 5.1); 12 h after the surgery—4.2 (3.86; 6.8); 7 days after the surgery—4.51 (3.03; 7.33); and 1 year after the surgery—0.12 (0.07; 0.295) mg/L. CRP levels in patients on bolus insulin therapy (Pump-) before the surgery—0.226 (0.097; 0.4); 1 h after the surgery—4.0 (1.7; 6.1); 12 h after the surgery—9.828 (8.255; 12.23); 7 days after the surgery—4.96 (2.46; 9.665); and 1 year after the surgery—0.17 (0.11; 0.4) mg/L.

**Figure 4 jcm-10-03399-f004:**
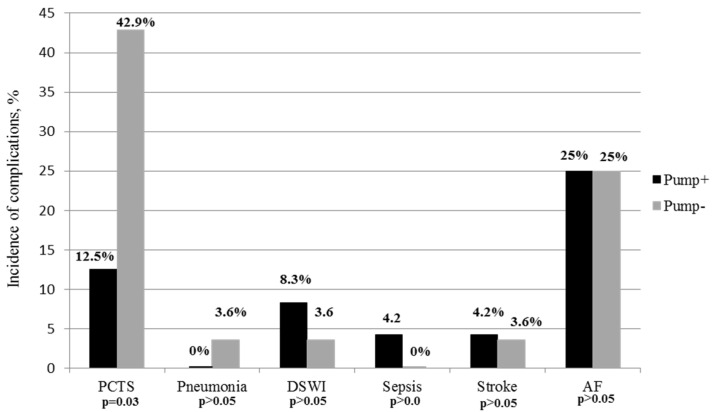
The incidence of complications in patients with DM2 in the early postoperative period, %. PCTS—postpericardiotomy syndrome; DSWI—deep sternal wound infection; AF-atrial fibrillations.

**Table 1 jcm-10-03399-t001:** Baseline characteristics of study patients.

Variable	Diabetes *n* = 52	No Diabetes *n* = 53	*p*
Age, years	63.25 (60.0; 68.0)	62.0 (59.0; 65.0)	0.05
Male, %	32.7	18.8	ns
BMI, kg/m^2^	30.45 (28.4; 32.9)	28.7 (26.8; 31.7)	0.03
History of CAD, years	4.0 (2.0; 11.0)	3.0 (1.0; 7.5)	ns
Postinfarction cardiosclerosis (PICS), %	63.5	52.8	ns
Hypertension, %	94.1	81.1	ns
Peripheral artery disease (PAD), %	5.8	7.6	ns
AF, %	8.3	6.1	ns
Creatinine, mmol/L	81.3 (73.9; 100.5)	84.3 (77.5; 103.0)	ns
GFR, ml/min	75.5 (63.0; 90.8)	85.0 (67.0; 92.0)	ns
СRP, mg/L	0.14 (0.07; 0.35)	0.26 (0.08; 0.64)	ns
Total cholesterol, mmol/L	4.3 (3.67; 4.9)	4.5 (3.82; 5.29)	ns
HDL cholesterol, mmol/L	2.35 (1.75; 2.87)	2.56 (1.94; 3.25)	ns
LDL cholesterol, mmol/L	1.08 (0.99; 1.13)	1.19 (1.0; 1.5)	ns
Venous blood glucose level, mmol/L	7.09 (5.5; 8.7)	5.4 (4.9; 6.2)	0.0005
HbA1c, %	6.8 (6.25; 7.66)	5.4 (5.18; 5.8)	0.000001
LVEF, %	60.0 (55.5; 64.0)	62 (57.9; 65.0)	ns
EDV	129 (107.3; 161.3)	130 (113.0; 147.5)	ns
IVS thickness	13.0 (12.0; 14.8)	13.0 (12.0; 14.0)	ns
On-Pump CABG, %	38.5	43.4	ns

BMI—body mass index; AF—atrial fibrillation; GFR—glomerular filtration rate; СRP-C—reactive protein; HDL—high-density lipoprotein; LDL—low-density lipoprotein; LVEF—left ventricular ejection fraction; EDV—end-diastolic volume; IVS—interventricular septum.

## Data Availability

Data are the property of the authors and are available upon request by contacting the corresponding author.

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
