# Peer review of "Should We Monitor Glucose and Biomarkers in Diabetics over Heart Surgery?"

_jcm, 2021, doi:10.3390/jcm10153399_

Round 1

Reviewer 1 Report

Thank you for submitting a revised version. I do not feel the authors have addressed all the concerns adequately:

  • I would still advise to include only patients with on pump surgery as the inflammatory response differs a lot in patients between on and off pump
  • More importantly as I read through the results I am still confused as to what the role of control population without diabetes is? Are the comparisons between the 24 patients in CGMS vs the 28+53 patients who got the hospital protocol of intermittent insulin bolus OR between the diabetic patients (n= 24 and 28 respectively) in the diabetic group. I could not clarify this in the read

I would redo the comparison with only diabetic patients and On pump surgery rather than listing them in the limitations

Author Response

Thank You very much for your work in revising our paper and your precious suggestions. Our detailed, point-by-point responses to the editorial and reviewer comments are given below, whereas the corresponding revisions are marked in colored text in the manuscript file.

Reviewer 2 Report

The manuscript has been improved.

The findings are not novel. The aim of the study still is unclear and the conclusions are not fully supported by the results. 

Author Response

Thank You very much for your work in revising our paper and your precious suggestions. Our detailed, point-by-point responses to the editorial and reviewer comments are given below, whereas the corresponding revisions are marked in colored text in the manuscript file.

This manuscript is a resubmission of an earlier submission. The following is a list of the peer review reports and author responses from that submission.

Round 1

Reviewer 1 Report

I have to raise several points:

  1. It is not known if the study is prospective or retrospective in nature.
  2. The aim of the study is unclear.
  3. The number of studied participants is low.
  4. The power analysis should be provided. Especially, if all-cause death was considered as the “primary outcome”.
  5. The analysis should be conducted with the aim of the study in the mind.
  6. What system of continuous glucose monitoring was used?
  7. Although the aim relates mainly to the continuous glucose monitoring the results deals mainly with insulin pump.
  8. The conclusion is not fully supported by the results.

The paper suffers from even more important limitations

Reviewer 2 Report

Thank you for the opportunity to review the manuscript " Should we monitor Glucose and Biomarkers in Diabetics Over heart Surgery?" by Gulokhova et al where they have compared post CABG glucose control with Insulin pumps vs standard intermittent dosage in diabetic patients and revealed better glycemic control with insulin pump in first 72 hours and also lower levels of markers of inflammation like Hs. CRP/ RANTES and also Leptin. There have been extensive studies in the past elucidating post CABG glucose control and outcomes and I do not feel this manuscript adds a whole lot to the available literature. Few suggestions for authors
1. Needs extensive language and grammar editing

2. The introduction is too vague and confused readers. Be specific to post cardiac surgery glucose control, level of control and outcomes

3. On pump and off pump patients have varied inflammatory responses. I would chose homogeneous population 

4. The results are very confusing. One way might be choosing only diabetic patients and also more tables rather than paragraphs.